# Global tree growth resilience to cold extremes following the Tambora volcanic eruption

Shan Gao [1], J. Julio Camarero [2], Flurin Babst [3,4] & Eryuan Liang [1] ✉

Although the global climate is warming, external forcing driven by explosive volcanic eruptions may still cause abrupt cooling. The 1809 and 1815 Tambora eruptions caused lasting cold extremes worldwide, providing a unique lens that allows us to investigate the magnitude of global forest resilience to and recovery from volcanic cooling. Here, we show that growth resilience inferred from tree-ring data was severely impacted by cooling in high latitudes and elevations: the average tree growth decreased substantially (up to 31.8%), especially in larch forests, and regional-scale probabilities of severe growth reduction (below −2$\sigma$) increased up to 1390%. The influence of the eruptions extended longer (beyond the year 1824) in mid- than in high-latitudes, presumably due to the combined impacts of cold and drought stress. As Tambora-size eruptions statistically occur every 200–400 years, assessing their influences on ecosystems can help humankind mitigate adverse impacts on natural resources through improved management, especially in high latitude and elevation regions.

Large volcanic eruptions are among the few natural phenomena that can force changes in the Earth's climate system[1–3]. The injection of sulfate aerosols can lead to a significant decrease in surface insolation and cause abrupt cooling at regional to global scales[4–6]. This cooling effect also dampens the global hydrological cycle and increases the likelihood of subsequent El Niño events, causing anomalously dry conditions across the monsoon regions[7,8]. Such extreme climate events severely impact forest resilience, reduce the provision of ecosystem services, and can even lead to societal collapses[9–12]. Therefore, a better understanding of how ecosystem resilience responds to abrupt global cooling will help us improve resource management in anticipation of future explosive volcanic eruptions.

The Mt. Tambora eruption in April 1815 was the largest explosive volcanic eruption recorded since AD 1400[5,13]. Model simulations reveal that the average global land surface temperature in June decreased by −1.9 °C in 1816, the so-called "year without a summer"[12]. Shortly before that, the undocumented 1809 eruption ranked as the third largest

eruption of its kind[14]. The two eruptions together made the 1810s the coldest decade during the past 500 years[15]. Hence, these explosive eruptions constitute ideal nature experiments for assessing the ecological consequences of lasting extreme cold events. Besides being a proxy for detecting historic volcanic signals and reconstructing climatic conditions as previous studies did[16–18], tree-ring width variability is also a direct indicator of ecological responses and resilience to such extreme events.

In this study, we assessed: (i) to what extent and (ii) for how long cooling after the 1809 and 1815 eruptions affected tree growth resilience using global tree-ring width data. Lasting growth reductions can lead to tree mortality[19] and decrease the productivity and stability of a forest ecosystem, which can foreshadow abrupt state shifts[20–22]. Both factors may indicate an abrupt change in forest growth resilience following anomalous climatic conditions. Therefore, to assess the magnitude of volcanically induced cold extremes, we calculated the change in radial growth and stability for the period 1815–1824 when

[1]State Key Laboratory of Tibetan Plateau Earth System, Environment and Resources (TPESER), Institute of Tibetan Plateau Research, Chinese Academy of Sciences, 100101 Beijing, China. [2]Instituto Pirenaico de Ecología (IPE-CSIC), 50059 Zaragoza, Spain. [3]School of Natural Resources and the Environment, University of Arizona, Tucson, AZ 85721, USA. [4]Laboratory of Tree-Ring Research, University of Arizona, Tucson, AZ 85721, USA. ✉e-mail: liangey@itpcas.ac.cn

volcanic forcing on climate was very pronounced[23,24]. We further applied superposed epoch analysis (SEA) to explore the timing and significance of growth departures from pre-eruption growth levels.

## Results

### Changes in the probability of severe growth reduction

The regional-scale probability of pronounced growth reduction (ring-width index (RWI) below $-1\sigma$) increased in almost all conifer forests across the selected regions (Supplementary Fig. 1) during the post-eruption (i.e., 1809–1824) relative to the pre-eruption period (i.e., 1759–1808). In boreal and alpine forests, the probability of growth extremes even increased up to more than tenfold. For instance, for evergreen conifer forests, the probability of tree growth dropping below $-2\sigma$ increased by 1207% in the Russian Arctic, 804% in the Tibetan Plateau, 1390% in northwestern North America, and 612% in the southern Andes. For deciduous conifer forests (i.e., larch), the probability of tree growth falling below $-2\sigma$ increased by 455% in the Russian Arctic and 1028% in western and central Europe. For broadleaf forests, the probability of pronounced growth reduction increased in central and eastern North America and the southern Andes and New Zealand, yet decreased in northern Europe, the Mediterranean region, and the west coast of the United States.

### Regional median growth anomalies

The volcanic eruptions exerted strong negative influences on tree growth in boreal and alpine forests, and relatively mild influences on temperate forests. The average growth during 5–10 years after the 1815 eruption, when the volcanic forcing on climate was the most pronounced, was severely reduced in boreal and alpine forests (Supplementary Fig. 2). Since persistent and significant growth reductions emerged irrespective of the chosen time window length, we only present the changes of growth resilience and stability for post-1815 eruption 8-years (i.e., 1815–1822) relative to the pre-1809 eruption decade (Fig. 1). Growth reductions were most severe (reduced by 31.8%) in the Russian Arctic. In addition, western and central Europe, the Mongolian and Tibetan Plateaus, northwestern North America, the west coast of the US, the Southwestern US (mainly mountainous regions in Arizona and New Mexico), central and eastern North America, and the southern Andes all experienced significant ($P < 0.05$) drops in growth for at least one forest functional type. Within each selected region, the intensity of resilience shifts also differed among forest types. For instance, the growth reductions of deciduous conifer forests (*Larix*) were stronger than those in evergreen needleleaf (*Pinus*, *Picea*) and deciduous broadleaf forests (e.g., *Betula*, *Quercus*, *Fagus*) on the Mongolian Plateau and in western and central Europe.

The lasting cold spell also caused regionally contrasting changes in growth stability (Supplementary Fig. 3, Fig. 1b). Growth stability was substantially reduced (42.0%) in the Southwestern US from 1815 to 1822 compared to the pre-eruption 10 years. Growth stability was also significantly ($P < 0.05$) reduced in the Russian Arctic, for larch forests in western and central Europe, for conifers along the west coast of the U.S. and in eastern North America, and for deciduous broadleaves (*Nothofagus*) in the southern Andes. By contrast, for evergreen needleleaf forests in northern Europe, in the Mediterranean region, and in New Zealand, as well as for deciduous broadleaf forests in western and central Europe, the growth stability significantly ($P < 0.05$) increased.

### Resilience shift trajectories

Regional growth-anomaly assessment revealed the trajectory of resilience changes after the two sequential eruptions (Fig. 2). A significant and continuous decrease in growth resilience occurred either just after the eruption (1809) year (i.e., in northwestern North America) or up to 3 years later. The strongest resilience decrease for a single year was found in the deciduous needleleaf forests from western and central Europe. In some boreal and alpine forests, a significant ($P < 0.05$)

resilience decrease lasted for over 10 years. The influence time of volcanic forcing was much longer than that of other disturbances during the pre-eruption 50 years (Supplementary Fig. 4) also caused severe growth reduction (RWI below $-1\sigma$) but with growth bouncing back either immediately, or within 2–3 years after the event. The longest influence times (over the year 1824) were found in the Mongolian and Tibetan Plateaus, the west coast of the US, mountainous regions of the Southwestern US, and central North America. The growth of evergreen needleleaf forests changed markedly in the Southwestern U.S.; there was no continuous significant reduction after the 1809 eruption, whereas, after the 1815 eruption, the growth was immediately enhanced for 3 years and then significantly reduced during 7 years. In the Mediterranean region and in eastern North America, several years of above-normal growth occurred, but only for evergreen needleleaf forests. In northern Europe, resilience was reduced for 3 years in evergreen needleleaf forests after the 1809 eruption, but growth was enhanced for 3 years after the 1815 eruption.

## Discussion

Through the study of tree rings, we found that tree growth rates in high latitude and elevation locations were severely impacted by the two sequential explosive volcanic events because forest growth was primarily limited by summer temperature in these regions. The probability of severe growth reduction after the eruptions increased substantially and the regional median growth reduction exceeded 30% for an 8-year long period (i.e., 1815–1822) in the Russian Arctic and in western and central Europe for deciduous needleleaf forests. This magnitude of regional growth anomaly exceeded growth reduction response to climatic extremes in other time periods that have led to substantial tree mortality[25]. The stability of tree growth also severely decreased in these regions. The combination of reduced growth levels and stability suggests a loss of growth resilience during the lasting cold spell. The growth stability decreased the most in the Southwestern US mountain forests, which may be due to the combined water and temperature limitation in this cold-dry environment. Cold conditions reduce evapotranspiration rates but also photosynthesis and growth rates.

At high latitudes and elevations, the growth of deciduous conifer forests (mainly larch) was less resilient to lasting cold extremes than evergreen needleleaf species (e.g., pine, spruce). The shorter leaf lifespan and higher wood density of deciduous conifers allow for a higher carbon assimilation rate than evergreen conifers under normal conditions[26,27]. During lasting cold events, however, these trees not only need to re-invest photosynthates into producing new leaves every year, but they also need more photosynthates to build dense woody tissues. Leaf lifespan seems to be directly associated with growth resilience in conifers[26] and may at least partly explain the differences we found.

The longest influence time of volcanic cooling on tree growth was found in mid-latitudes in the Northern Hemisphere; significantly reduced growth resilience extended over 1824, that is, 10 years after the Tambora eruption. On the Mongolian and Tibetan Plateaus where vegetation growth is limited by both low growing-season temperature and precipitation, resilience decreased following the 1809 eruption and lasted until several years after the 1815 eruption; along the west coast of the US, the Southwestern US mountainous regions, and central North America, vegetation growth is more limited by water availability than by temperature. Accordingly, we observed a rapid return of decreased growth to normal after the 1809 eruption, but a lasting resilience decreased followed the more explosive 1815 eruption. The prolonged response of tree radial growth in temperature-limited regions indicates stronger memory effects to extreme cold compared to non-temperature-limited regions due to their more energy-driven growth processes[28–32]. A decline in precipitation due to volcanic cooling may have had further impacts on growth in mid-latitude regions,

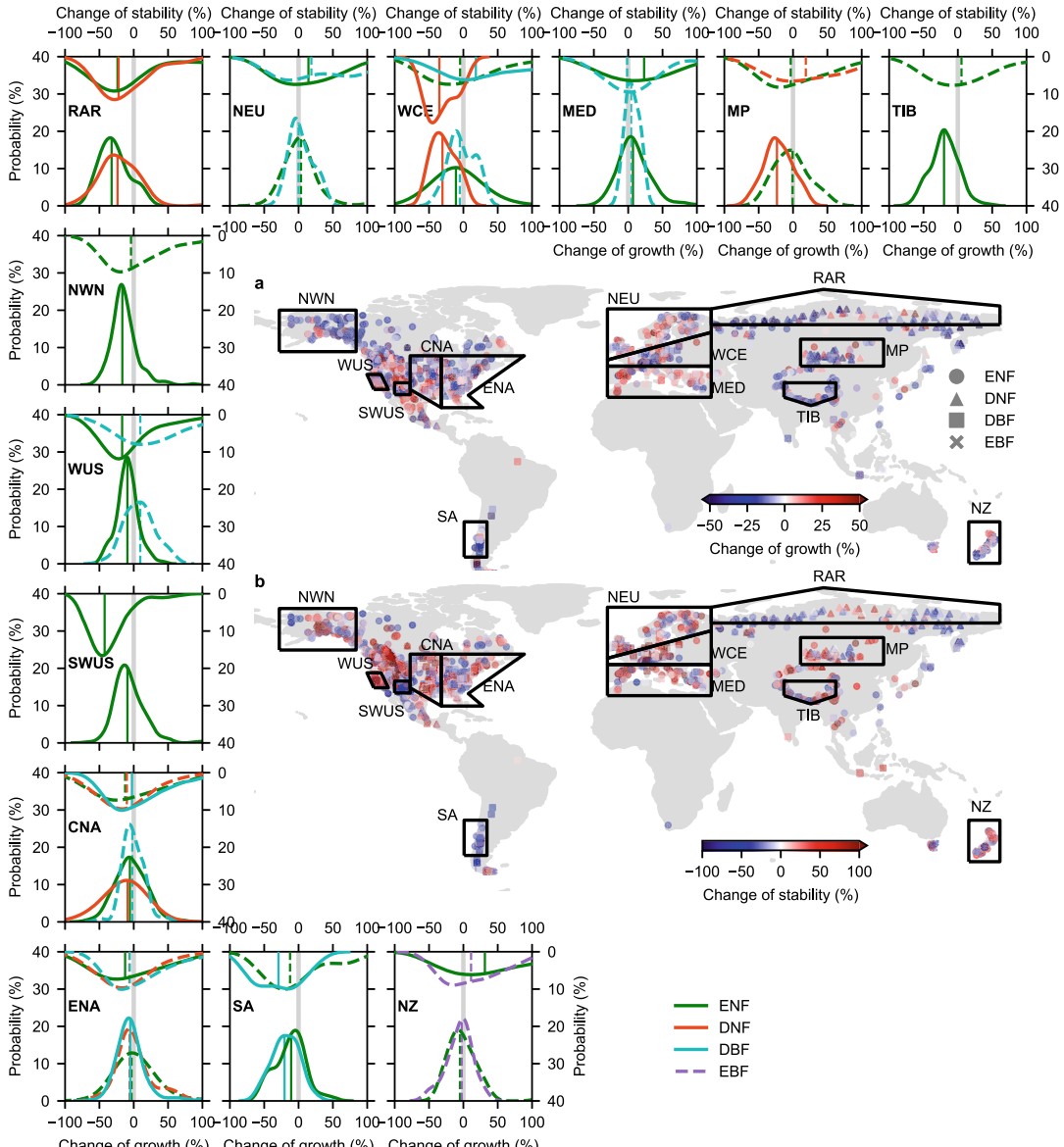

**Fig. 1 | Regional tree growth response to the 1815 Tambora eruption.** Colors of markers on the geographic maps represent the site-level change in growth (**a**) and stability (**b**) during a fixed period (1815–1822) compared to their pre-eruption period of 1789–1808 for each tree-ring chronology. Statistical plots show the probability distribution of the change of growth resilience ($R_f$, presented in bottom and left coordinate axes) and stability ($S_f$, presented in top and right coordinate axes) for evergreen needleleaf forests (ENF, green lines), deciduous needleleaf forests (DNF, orange red lines), deciduous broadleaf forests (DBF, cyan lines) and evergreen broadleaf forests (EBF, purple lines) within selected regions. Solid/dashed curves represent significant/non-significant differences (at the

$\alpha = 0.05$ level) between the mean growth and stability of the fixed period and pre-eruption period (i.e., 1799–1808) estimated using the two-sided Wilcoxon test. RAR, NEU, WCE, MED, MP, TIB, NWN, WUS, SWUS, CNA, ENA, SA and NZ represent Russian-Arctic, northern Europe, western and central Europe, the Mediterranean region, the Mongolian Plateau, the Tibetan Plateau, northwestern North America, the west coast of the US, Southwestern US (mainly Arizona and New Mexico mountainous region), central North America, eastern North America, the southern Andes and New Zealand, respectively. Vertical lines represent the median of the distributions of $R_f$ and $S_f$ for each forest type within a given region.

delaying the return of tree growth to pre-eruption levels. For example, a significant reduction in spring and summer precipitation happened immediately after large tropical volcanic eruptions and may last several years in the southern Tibetan Plateau, which reduced the growth of trees[33,34]. The positive 3-year lagged response of resilience observed for most regions may be due to diffuse light fertilization on tree photosynthesis, which can benefit carbon allocation to growth even when other environmental factors are limiting[35].

The large-magnitude and long-term reduction in growth found in trees is in accordance with historical records that reported harvest failure and food shortage in central Europe, North America, and China following these volcanic eruptions[10,36,37]. Nevertheless, not all regions

experienced a severe reduction in growth resilience. Though located in high latitudes, the change of tree growth levels in northern Europe was mild, and regional median growth stability was enhanced, which may be related to the North Atlantic Current that transports warm waters to high latitudes. In the coniferous forests in the Mediterranean region and eastern North America, cool summers enhanced growth levels, as well as growth stability, presumably due to the associated decrease in evapotranspiration rates and increased water availability for tree growth. Forests in New Zealand did not show significant growth reduction. This may be because the cooling effects were less strong in the Southern Hemisphere[38], and surrounding ocean areas buffered the cooling effects.

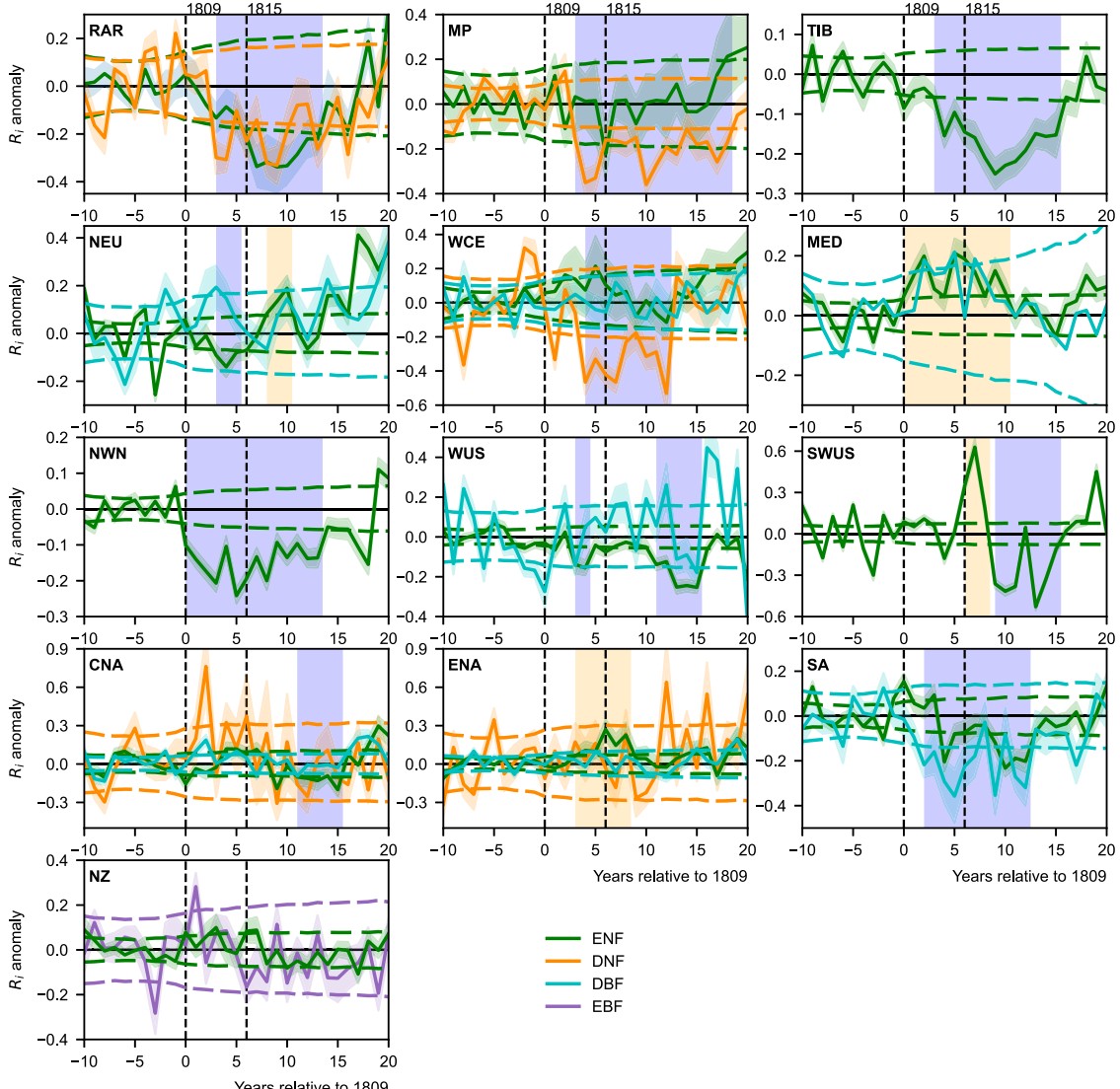

**Fig. 2 | Regional growth resilience ($R_i$) anomaly after the 1809 and 1815 volcanic eruptions within selected regions.** The expected mean (solid lines) is bounded by the 95% confidence intervals (shaded horizontal boundary) for evergreen needle-leaf forests (ENF, green lines), deciduous needleleaf forests (DNF, orange lines) and deciduous broadleaf forests (DBF, cyan lines). The horizontal dashed lines indicate the threshold required for epochal anomalies to be statistically significant at the $\alpha = 0.05$ level for different forest types. The purple and yellow shading marks the continuous periods when growth anomalies for either forest type were significant.

RAR, NEU, WCE, MED, MP, TIB, NWN, WUS, SWUS, CNA, ENA, SA and NZ represent Russian-Arctic, northern Europe, western and central Europe, the Mediterranean region, the Mongolian Plateau, the Tibetan Plateau, northwestern North America, the west coast of the US, Southwestern US (mainly Arizona and New Mexico mountainous region), central North America, eastern North America, the southern Andes and New Zealand, respectively. Vertical dashed lines marked out the eruption years.

In summary, this study evidenced the changes in forest growth and stability after the 1809 and 1815 volcanic eruptions across forests worldwide. The volcanic eruptions exerted strong negative influences on tree growth in boreal and alpine forests, and relatively mild influences for temperate forests. Both growth levels and stability dropped significantly in the Russian Arctic and in western and central Europe for deciduous larch forests, and growth stability decreased the most in the Southwestern US mountain forests. The longest continuous influence time appeared in the Mongolian and Tibetan Plateaus, and the influence time extended longer in mid-latitude regions than in high-latitude regions. Although the current and projected trajectory of climate change is toward warming, external forcing to climate-driven by explosive volcanic eruptions may still cause sudden drops in global temperature. Probabilistic eruption forecasting suggested that Tambora-size volcanic eruptions may occur every 200–400 years[39].

These forecasted eruptions have also been predicted to lead to similar or even stronger global cooling effects as/then in the 19th century and may amplify El Niño-like effects on some monsoon regions[40,41] under climate change. Decreases in forest growth and stability indicate an overall decline in ecosystem services to human society, which leads to an increased risk of social unrest. Therefore, assessing ecosystem resilience as done in this study can help mankind to better prepare for and cope with adverse impacts on natural resources of global cold spells by improving ecosystem management, especially in regions located at high latitudes and high elevations.

## Methods
### Tree-ring width chronologies
Global raw tree-ring width chronologies were collected from three sources, the International Tree-Ring Data Bank (ITRDB) (https://www.

ncei.noaa.gov/pub/data/paleo/treering/measurements/), the GenTree dendroecological collection[42] (https://doi.org/10.6084/m9.figshare.c.4561037.v1) and the collection from the tree-ring group of the Institute of Tibetan Plateau Research, Chinese Academy of Sciences (ITPCAS)[43] (https://doi.org/10.11888/Terre.tpdc.300576). We retained only tree-ring width chronologies that date back to at least 1759, so that as Tambora erupted in 1815, the responses of those forests were minimally affected by tree-age effects. As a result, a total of 3304 tree-ring width chronologies were retained for further analyses. We then transformed the raw tree-ring width data (RW) into ring-width indices (RWI) by removing long-term trends caused by aging and increasing trunk diameter. Different detrending techniques may cause uncertainties when deriving the influence time of an extreme event. Since the two major consecutive eruptions were recorded to have long-term effects on ecosystems, we chose to use the most conservative standardization technique of fitting negative exponential curves which follows a model of tree growth to detrend tree-level series using the dplR package[44] (version 1.7.1) in R[45]. Site-level, standard chronologies of RWI were averaged using a bi-weight robust mean.

For comparisons, a 30-year cubic smoothing spline was also used for detrending in order to verify the robustness of the derived influence time of volcanic forcing on tree growth against our choice of detrending method. This detrending method following an empirical model, removes 50% of variance at 30 years and leaves 99% of variance at 9.5 years[46]. Although the lasting time showed a few years difference compared to detrending by fitting negative exponential curves, it still showed that the influence time can extend longer in mid-latitude regions than in high-latitude regions.

## Calculation of forest growth resilience

In this study, both changes in growth resilience and stability were estimated. We used the shift of RWI from the pre-eruption state to indicate growth resilience. We calculated the change in growth resilience for a set of window lengths of 5–10 years after the 1815 eruption ($R_f$, Eq. (1)). To examine the process of resilience shifts after the eruptions, we also calculated annual growth resilience ($R_i$) from 1809 to 1829 (Eq. (2)). Stability was defined as the ratio of the mean growth to its variance for a certain period. The change of stability was calculated as the change of the ratio of mean growth to its variance ($\sigma$) for a set of window lengths of 5–10 years after the 1815 eruption (Eq. (3)).

$$R_f = \frac{\overline{RWI_{1815-a}} - \overline{RWI_{1799-1808}}}{\overline{RWI_{1799-1808}}} \quad (1)$$

$$R_f = \frac{\overline{RWI_i}}{\overline{RWI_{1799-1808}}} \quad (2)$$

$$S_f = \frac{\left(\frac{\overline{RWI_{1815-a}}}{\sigma_{1815-a}} - \frac{\overline{RWI_{1799-1808}}}{\sigma_{1799-1808}}\right)}{\frac{\overline{RWI_{1799-1808}}}{\sigma_{1799-1808}}} \quad (3)$$

where $\overline{RWI_{1799-1808}}$ is the pre-eruption (year 1799–1808) averaged RWI; $\overline{RWI_{1815-a}}$ is the average RWI for a fixed period of 5–10 years after the 1815 eruption, $a$ is ranged from 1819 to 1824; $\overline{RWI_i}$ is the RWI for year $i$, and $i$ is the year ranging from 1809 to 1829.

## Regional growth-anomaly assessment on growth resilience

A modified double bootstrap superposed epoch analysis (SEA) can isolate pulse disturbance signals against a large amount of background noise through composite averaging[47]. In this study, we modified this approach to evaluate the probability of the association between discrete volcanic eruptions and subsequent growth resilience ($R_i$) shifts on regional scales. We selected 13 regions on the basis of the spatial patterns of the site-level $R_f$ and climatological consistency to examine

the regional response of growth resilience to volcanic eruptions. The definition of these regions referenced to IPCC AR6 climate reference regions[48]. The number of chronologies for each forest functional type within each study region is presented in Supplementary Table 1. We established a regional series by joining a set of site-level $R_i$ series within a selected region and applied the SEA to the joint regional series, comparing the 10 years before and 20 years after the first eruption year (1809). We made 1000 unique composite matrices using random combinations of approximately half the total number of the regional key events without replacement and determined the statistical significance of the growth response by comparing its probability distribution to that generated from 10,000 iterations of pseudo-composite matrices.

The SEA analysis was also used to assess the baseline of growth resilience in the absence of volcanic impacts for the period of 1759–1808 as shown in Supplementary Fig. 4. For each functional type of forest within the study regions, we identified the year when the largest number of tree-ring sites showed homogeneous growth decrease (i.e., RWI < −1$\sigma$). We then used the RWI chronologies with homogeneous growth decrease to establish the regional series by joining them within the region and applied the SEA to the joint regional series, comparing the 10 years before and 20 years after the identified year.

## Reporting summary
Further information on research design is available in the Nature Portfolio Reporting Summary linked to this article.

## Data availability
The ITRDB tree-ring width data is obtained from https://www.ncei.noaa.gov/pub/data/paleo/treering/measurements. The GenTree dendroecological collection tree-ring width data is obtained from https://doi.org/10.6084/m9.figshare.c.4561037.v1. Tree-ring width data from the ITPCAS tree-ring group are available from https://doi.org/10.11888/Terre.tpdc.300576.

## Code availability
Statistical analysis in this study was performed with publicly available packages in R (version 3.6.2), and the custom code for the analysis of the data is available from https://doi.org/10.11888/Terre.tpdc.300576.

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

## Acknowledgements

We acknowledge all contributors to the International Tree-Ring Data Bank and the GenTree Dendroecological Collection for providing tree-ring data. This study was supported by the National Natural Science Foundation of China (42030508, 41988101, 41907387), the Second Tibetan Plateau Scientific Expedition and Research Program (2019QZKK0301), the Science and Technology Major Project of Tibetan Autonomous Region of China (XZ202201ZD0005G02).

## Author contributions

E.L. proposed the idea, and S.G. performed the analysis and drafted the manuscript. S.G., J.J.C., F.B. and E.L. discussed the results and improved the manuscript.

## Competing interests

The authors declare no competing interests.
