## [Peer Review File · Nature Communications]

Reviewers' Comments:

Reviewer #1:

Remarks to the Author:

This study investigates tree growth resilience during 1809 and 1815 Tambora eruptions by using tree-ring width data worldwide and the findings highlight the significant reduction in growth rates and stability in high latitude and elevation areas, providing insights into the potential consequences of future volcanic events. Overall, I think the paper have a valuable contribution to the understanding of ecological response to volcanic cooling. However, I also have a few suggestions or comments. I suggest publishing the paper after minor revisions.

One suggestion is to consider how tree ecological resilience varies globally in the absence of volcanic impacts, in order to better explain the effects of extreme cold. For instance, exploring the recovery period in different regions when facing similar levels of disturbance can provide insights into the baseline resilience of forests.

Additionally, I have a question. The volcanic eruption in 1809 may not have affected forests globally, leading to disturbances in some regions while others remained unaffected. Therefore, considering this, did the global forest response after 1815 show any influence from the previous disturbance? Perhaps the authors can consider adding content about ecological memory in the discussion section.

Minor comments:

1. Line 15: There might be a missing word in the sentence "little is known a global pattern..."

2. Please provide more quantitative information (such as the extent of the pronounced drops) and more details of the observed global patterns in Results section.

Reviewer #2:

Remarks to the Author:

General comments

The manuscript "Global tree growth resilience to cold extremes" by Gao et al. is a timely example showing a global synchronic response of trees on cooling impact of Tambora eruptions at 1809 and 1815. Taking into account increasing global climate "nervousness" when heat waves can be rapidly changed by short cold extremes even during vegetation season, little is known a tree growth response to extreme cooling. Using the worldwide distributed tree-ring width data from the widely used ITRDB and very simple statistical approaches the authors show what the tree-ring response of different tree species could be for the most of forest ecosystems in the planet. The authors refer to this response as «resilience», which translates into a global short-term growth depression followed by a 7-10 year recovery highlighted especially for deciduous conifer species.

The paper is well structured. The introduction provided a good overview of the background information and the pertinent literature, and it demonstrated the need for the current study. The most of the results were confirmed statistically and well presented visually. Based on it the authors provided a satisfactory discussion where it was shown, as an example, deciduous conifer trees were less resilient to cold extremes than evergreen needleleaf species. The authors also considered a changing role of principle growth-limiting factors before and after the cold impact in global scale.

But there are two serious issues that should be considered in the MS.

In this study the authors used two time windows. The first window (left end – 1799-1808 (10 years before the eruption); right end – 1815-1722 (the 8-years post-eruptions period)) (see equations (1)-(3)) is used to estimate the "change of growth" and the "change of stability" (see the Fig.1). The second asymmetric 30-years (10 years before 1809; and 20-years after) window is applied in Superposed Epoch Analysis (see the Fig.2). I am sure that changing the "window" length

in both cases may affect the final results. I would recommend that the authors show the stability of the results for various asymmetric window lengths or describe the criteria for selecting window lengths.

The Figure 1 is very important, the authors showed the distributions of "change in growth" and "change in stability" during 1815-1822. for all regions and forest types relative to median growth. Is it possible to show that those shifts are statistically significant relative to Zero? Could the authors apply some statistical criteria to show significant or insignificant shifts? What does "median" mean (see the Fig.1) and what is a time period for which those medians were obtained? 1799-1808 or 1815-1822 or 1759 till the end of tree-ring chronology (see the Line 184) or others? Clarification is needed.

I would recommend to publish the MS after major revision.

Specific comments

Lines 188-190: Why were negative exponential curves selected as a detrended approach taking into account that the simple exponent can be a reason of bias for the end of individual tree-ring series (Melvin, 2004)? Clarification is needed.

Lines 194-202: How was the window length (left end – 1799-1808; right end – 1815-1722) selected?

Equation 2: I would recommend the Authors to indicate that the R_i was used in Superposed Epoch Analysis and in Fig.2

Lines 215-217: Why was New Zealand not selected as a region for analysis taking into account its closest neighborhood with Tambora.

Lines 215-217: Why was the most richest dendroregion (the central and southern West Coast of US) represented just by the southern part of the Colorado Plateau?

The reference list should be checked. The reference number 38 in the main text is not presented in the reference list.

Ms. NCOMMS-23-20595A: Point-by-point response to comments of

Reviewer 1

[Comments] *This study investigates tree growth resilience during 1809 and 1815 Tambora eruptions by using tree-ring width data worldwide and the findings highlight the significant reduction in growth rates and stability in high latitude and elevation areas, providing insights into the potential consequences of future volcanic events. Overall, I think the paper have a valuable contribution to the understanding of ecological response to volcanic cooling. However, I also have a few suggestions or comments. I suggest publishing the paper after minor revisions.*

[Responses] We appreciate the reviewer's positive assessment of the study. We also very much appreciate the suggestions that the reviewer brought up, which have helped us to further refine this manuscript.

[Comments] *One suggestion is to consider how tree ecological resilience varies globally in the absence of volcanic impacts, in order to better explain the effects of extreme cold. For instance, exploring the recovery period in different regions when facing similar levels of disturbance can provide insights into the baseline resilience of forests.*

[Responses] We agree that providing a resilience baseline would highlight the severity of volcanic impacts on global tree growth. Following the suggestion, we have now added Supplementary Figures 1 and 4 to describe the general extreme responses of tree growth. We have also added respective descriptions in the Results section (please see below).

In the Results section, we added [The regional-scale probability of pronounced growth reduction (ring-width index (RWI) below -1σ) increased in almost all conifer forests across the selected regions (Supplementary Figure 1) during the post-eruption (i.e., 1809 to 1824) relative to the pre-eruption period (i.e., 1759-1808). In boreal and alpine forests, the probability of growth extremes even increased up to more than tenfold. For instance, for evergreen conifer forests, the probability of tree growth dropping below -2σ increased by 1207% in the Russian Arctic, 804% in the Tibetan Plateau, 1390% in northwestern North America, and 612% in the southern Andes. For deciduous conifer forests (i.e., larch), the probability of tree growth falling below -2σ increased by 455% in in the Russian Arctic and 1028% in western and central Europe. For broadleaf forests, the probability of pronounced growth reduction increased in central and eastern North America and the southern Andes and New Zealand, yet decreased in northern Europe, the Mediterranean region, and the west coast of the United States.

...In some boreal and alpine forests, significant ($P < 0.05$) resilience decrease lasted for over 10 years. The influence time of volcanic forcing was much longer than that of other disturbances during the pre-eruption 50 years (Supplementary Figure 4) that also caused severe growth reduction (RWI below -1σ) but with growth bouncing back either immediately,

or within 2 to 3 years after the event...]

Supplementary Figures are shown below:

Supplementary Figure 1 | Change of growth extremes after 1809 and 1815 volcanic eruptions (1809-1824) compared to 1759-1808. The percentages in each panel showed the increased ratio of the percentage of RWI below $-1/2$ standard deviation (σ) in 1809-1829 relative to 1759-1808. ENF, DNF, DBF and EBF represent evergreen needleleaf forests, deciduous needleleaf forests, deciduous broadleaf forests and evergreen broadleaf forests, respectively. RAR, NEU, WCE, MED, MP, TIB, NWN, WUS, AM, CNA, ENA, SA and NZ represent Russian-Arctic, northern Europe, western and central Europe, the Mediterranean region, the Mongolian Plateau, the Tibetan Plateau, northwestern North America, the west coast of US, the Southwestern U.S., central North America, eastern North America, the southern Andes and New Zealand, respectively.

Supplementary Figure 4 | Tree growth recovery baseline without volcanic forcing for each region in 1759 to 1808. The expected mean (solid lines) is bounded by the 95% confidence intervals (shaded horizontal boundary) for evergreen needleleaf forests (ENF, green lines), deciduous needleleaf forests (DNF, orange red lines) and deciduous broadleaf forests (DBF, cyan lines). The horizontal dashed lines indicate the threshold required for epochal anomalies to be statistically significant at the $\alpha = 0.05$ level. RAR, NEU, WCE, MED, MP, TIB, NWN, WUS, SWUS, CNA, ENA, SA and NZ represent Russian-Arctic, northern Europe, western and central Europe, the Mediterranean region, the Mongolian Plateau, the Tibetan Plateau, northwestern North America, the west coast of US, the Southwestern U.S., central North America, eastern North America, the southern Andes and New Zealand, respectively. Vertical dashed lines marked out the year when largest number of sites showed homogeneous growth decrease (i.e., RWI below -1σ).

[Comments] *Additionally, I have a question. The volcanic eruption in 1809 may not have affected forests globally, leading to disturbances in some regions while others remained unaffected. Therefore, considering this, did the global forest response after 1815 show any influence from the previous disturbance? Perhaps the authors can consider adding content about ecological memory in the discussion section.*

[Responses] Yes, mainly boreal and alpine forests show significant resilience decreases following the 1809 eruption. Following the reviewer's suggestion, we have now added related discussion in the Discussion section: [The longest influence time of volcanic forcing on tree growth were found in mid-latitudes in the Northern Hemisphere; significantly reduced growth resilience can extend to 1824, that is 9 years after the Tambora eruption. On the Mongolian and Tibetan Plateaus, where vegetation growth is limited by both low growing-season temperature and precipitation, resilience decreased followed the 1809 eruption and last till several years after the 1815 eruption; in the west coast of US, Arizona mountainous region, central North America, vegetation growth is more limited by water availability rather than temperature, there appeared rapid return of decreased resilience after the 1809 eruption, and continuous resilience decreased followed the more explosive 1815 eruption. The temporal extended response of tree radial growth in temperature-limited regions indicates stronger memory effects to extreme coldness than non-temperature limited regions due to their more temperature-linked growth process (Esper et al., 2015; Rossi et al., 2016; Li, et al., 2017; Mu et al., 2023).]

[Comments] *Minor comments:*

1. Line 15: There might be a missing word in the sentence "little is known a global pattern..."

[Responses] Following the comment, this sentence could be revised into: [However, little is known on global patterns of tree growth resilience to cold extremes that may happen in a warmer climate.] In the revised version, we have rewritten a more concise abstract to meet the word requirement: [Although the global climate is warming and drying, external forcing driven by explosive volcanic eruptions may still cause abrupt cooling. The 1809 and 1815 Tambora eruptions caused lasting cold extremes worldwide, providing a unique lens that allowed us to investigate the magnitude of global forest resilience to and recovery from volcanic cooling. Growth resilience inferred from tree-ring data was severely impacted by cooling in high latitudes and elevations: regional-scale probabilities of severe growth reduction (below -2σ) increased up to 1390%, the average tree growth decreased substantially (up to 31.8%), especially in larch forests. The influence time extended longer (over the year 1824) in mid- than in high-latitudes, presumably due to the combined impacts of cold and drought stress., presumably due to the combined impacts of cold and drought stress. As Tambora-size eruptions statistically occur every 200-400 years, assessing their influences on ecosystems can help mankind mitigate adverse impacts on natural resources through improved management, especially in high latitude and elevation regions.]

[Comments] *2. Please provide more quantitative information (such as the extent of the pronounced drops) and more details of the observed global patterns in Results section.*

[Responses] Following the suggestion, we have now added statistical significance tests to show the extent of resilience drops and added more quantitative information and details of the observed global patterns in the Results section: [The regional-scale probability of pronounced growth reduction (ring-width index (RWI) below -1σ) increased in almost all conifer forests across the selected regions (Supplementary Figure 1) during the post-eruption (i.e., 1809 to 1824) relative to the pre-eruption period (i.e., 1759-1808). In boreal and alpine forests, the probability of growth extremes even increased up to more than tenfold. For instance, for evergreen conifer forests, the probability of tree growth dropping below -2σ increased by 1207% in the Russian Arctic, 804% in the Tibetan Plateau, 1390% in northwestern North America, and 612% in the southern Andes. For deciduous conifer forests (i.e., larch), the probability of tree growth falling below -2σ increased by 455% in the Russian Arctic and 1028% in western and central Europe. For broadleaf forests, the probability of pronounced growth reduction increased in central and eastern North America and the southern Andes and New Zealand, yet decreased in northern Europe, the Mediterranean region, and the west coast of the United States.

...Growth reductions were most severe (reduced by 31.8%) in the Russian Arctic. In addition, western and central Europe, the Mongolian and Tibetan Plateaus, northwestern North America, the west coast of the U.S., the Southwestern U.S. (mainly mountainous regions in Arizona and New Mexico), central and eastern North America, and the southern Andes all experienced significant ($P < 0.05$) drops in growth for at least one forest functional type.

...Growth stability was substantially reduced (42.0%) in the Southwestern U.S. from 1815 to 1822 compared to the pre-eruption 10 years. Growth stability was also significantly ($P < 0.05$) reduced in the Russian Arctic, for larch forests in western and central Europe, for conifers along the west coast of the U.S. and in eastern North America, and for deciduous broadleaves (*Nothofagus*) in the southern Andes. By contrast, for evergreen needleleaf forests in northern Europe, in the Mediterranean region, and in New Zealand, as well as for deciduous broadleaf forests in western and central Europe, the growth stability significantly ($P < 0.05$) increased.]

The revised Figure 1 is shown below:

Figure 1. Regional tree growth response to the 1815 Tambora eruption. Colors of markers on the geographic maps represent the change in growth (A) and stability (B) during a fixed period (1815 to 1822) comparing to their pre-eruption period of 1789 to 1808 for each tree-ring chronology. Statistical plots show the probability distribution of the change of growth resilience (R_f , presented in bottom and left coordinate axes) and stability (S_f , presented in top and right coordinate axes) for evergreen needleleaf forests (ENF, green lines), deciduous needleleaf forests (DNF, orange red lines), deciduous broadleaf forests (DBF, cyan lines) and evergreen broadleaf forests (EBF, purple lines) within selected regions. Solid/dashed curves represent significant/non-significant difference ($P < 0.05$) between the mean growth and stability of the fixed period and pre-eruption period (i.e., 1799-1808) estimated using the Wilcoxon test. RAR, NEU, WCE, MED, MP, TIB, NWN, WUS, SWUS, CNA, ENA, SA and NZ represent Russian-Arctic, northern Europe, western and central Europe, the Mediterranean region, the Mongolian Plateau, the Tibetan Plateau, northwestern North America, the west coast of US, the Southwestern U.S., central North America, eastern North America, the southern Andes and New Zealand, respectively. Vertical dashed lines represent the median of

the distributions of R_f and S_f for each forest type within a given region.

References

- Esper, J. et al. Signals and memory in tree-ring width and density data. *Dendrochronologia*. **35**, 62-70 (2015).
- Li, X. et al. Critical minimum temperature limits xylogenesis and maintains treelines on the southeastern Tibetan Plateau. *Sci. Bull.* **62**, 804-812 (2017).
- Mu, Y. et al. Tree-ring evidence of ecological stress memory. *P. Roy. Soc. B-Biol. Sci.* **289**, 20221850 (2022).
- Rossi, S. et al. Pattern of xylem phenology in conifers of cold ecosystems at the Northern Hemisphere. *Glob. Change. Biol.* **22**, 3804-3813 (2016).

Ms. NCOMMS-23-20595A: Point-by-point response to comments of

Reviewer 2

[Comments] *General comments*

The manuscript “Global tree growth resilience to cold extremes” by Gao et al. is a timely example showing a global synchronic response of trees on cooling impact of Tambora eruptions at 1809 and 1815. Taking into account increasing global climate “nervousness” when heat waves can be rapidly changed by short cold extremes even during vegetation season, little is known a tree growth response to extreme cooling. Using the worldwide distributed tree-ring width data from the widely used ITRDB and very simple statistical approaches the authors show what the tree-ring response of different tree species could be for the most of forest ecosystems in the planet. The authors refer to this response as «resilience», which translates into a global short-term growth depression followed by a 7-10 year recovery highlighted especially for deciduous conifer species.

The paper is well structured. The introduction provided a good overview of the background information and the pertinent literature, and it demonstrated the need for the current study. The most of the results were confirmed statistically and well presented visually. Based on it the authors provided a satisfactory discussion where it was shown, as an example, deciduous conifer trees were less resilient to cold extremes than evergreen needleleaf species. The authors also considered a changing role of principle growth-limiting factors before and after the cold impact in global scale. I would recommend to publish the MS after major revision.

[Responses] We appreciate the positive evaluation of the reviewer regarding the importance and interest of our study, the robustness of our methodology and results, as well as the structure of writing. We also very much appreciate his/her insightful comments, which have helped us to improve the manuscript considerably.

[Comments] *But there are two serious issues that should be considered in the MS.*

In this study the authors used two time windows. The first window (left end – 1799-1808 (10 years before the eruption); right end – 1815-1722 (the 8-years post-eruptions period)) (see equations (1)-(3)) is used to estimate the “change of growth” and the “change of stability” (see the Fig.1). The second asymmetric 30-years (10 years before 1809; and 20-years after) window is applied in Superposed Epoch Analysis (see the Fig.2). I am sure that changing the “window” length in both cases may affect the final results. I would recommend that the authors show the stability of the results for various asymmetric window lengths or describe the criteria for selecting window lengths.

[Responses] Yes, the reviewer is correct that the results may slightly vary using the mean growth and stability in different time windows. Following their suggestion, we have added Supplementary Figures 2 and 3 to demonstrate the robustness of our results to the choice of time window (please see below). Reassuringly, the change of time window did not affect the overall conclusions of this study. Many studies that used SEA to analyze the influences of natural hazard (e.g., flood, volcanism) on tree growth chose the pre- and post-event periods to last around 3-5 years (e.g., Rao et al., 2020; Dee et al., 2020; Liu et al., 2022). The Tambora

eruption is the largest explosive volcanic eruption recorded for the past 600 years and tree-ring evidence (including ours) suggests its impacts could last for more than 10 years in some regions (e.g., the Mongolian and the Tibetan Plateau). Therefore, we have extended the temporal window to 20 years after the 1809 eruption to properly display the duration of the volcanic influences. Changing the time windows did not affect the result of the SEA analysis, it only affects the window length displaying the results.

Supplementary Figure 2 | Box plot of averaged ring-width index (\overline{RWI}) for different pre- and post-eruption periods in the 13 regions. Pre-eruption 20-, 10- and 5-years represent the period of 1789-1808, 1799-1808 and 1804-1808, respectively; post-eruption 5-, 6-, 7-, 8-, 9- and 10-years represent the period of 1815 to 1819, 1820, 1821, 1822, 1823 and 1824, respectively. In each panel, color filled boxes represent significant difference ($P < 0.05$) of tree growth relative to the pre-eruption 10-years (shaded in grey) estimated with the Wilcoxon test. ENF, DNF, DBF and EBF represent evergreen needleleaf forests, deciduous needleleaf forests, deciduous broadleaf forests and evergreen broadleaf forests, respectively. RAR, NEU, WCE, MED, MP, TIB, NWN, WUS, SWUS, CNA, ENA, SA and NZ represent Russian-Arctic, northern Europe, western and central Europe, the Mediterranean region, the Mongolian Plateau, the Tibetan Plateau, northwestern North America, the west coast of US,

the Southwestern U.S., central North America, eastern North America, the southern Andes and New Zealand, respectively.

Supplementary Figure 3 | Box plot of tree growth stability for different pre- and post-eruption periods in the 13 regions. Pre-eruption 20-, 10- and 5-years represent the period of 1789-1808, 1799-1808 and 1804-1808, respectively; post-eruption 5-, 6-, 7-, 8-, 9- and 10-years represent the period of 1815 to 1819, 1820, 1821, 1822, 1823 and 1824, respectively. In each panel, color filled boxes represent significant difference ($P < 0.05$) of tree growth stability relative to the pre-eruption 10-years (shaded in grey) estimated with the Wilcoxon test. ENF, DNF, DBF and EBF represent evergreen needleleaf forests, deciduous needleleaf forests, deciduous broadleaf forests and evergreen broadleaf forests, respectively. RAR, NEU, WCE, MED, MP, TIB, NWN, WUS, SWUS, CNA, ENA, SA and NZ represent Russian-Arctic, northern Europe, western and central Europe, the Mediterranean region, the Mongolian Plateau, the Tibetan Plateau, northwestern North America, the west coast of US, the Southwestern U.S., central North America, eastern North America, the southern Andes and New Zealand, respectively.]

[Comments] The Figure 1 is very important, the authors showed the distributions of "change in

growth" and "change in stability" during 1815-1822. for all regions and forest types relative to median growth. Is it possible to show that those shifts are statistically significant relative to Zero? Could the authors apply some statistical criteria to show significant or insignificant shifts? What does "median" mean (see the Fig.1) and what is a time period for which those medians were obtained? 1799-1808 or 1815-1822 or 1759 till the end of tree-ring chronology (see the Line 184) or others? Clarification is needed.

[Responses] We agree that showing the significant shifts via statistical test is very important to demonstrate the extremeness of volcanic impacts. Following the comment, we have now used solid and dashed curves to represent the significant (at 0.05 level) and non-significant shifts of post-volcanic growth compared to the pre-eruption period. Since the two sets of ring-width indices may not be always normally distributed, we used the non-parametric Wilcoxon test to test their difference. The significance differences of mean growth and stability were also marked for different post-volcanic time windows in Supplementary Figure 2 and 3.

In Figure 1, the "median" means the median values for the distributions of site-level change of growth resilience (R_f) and stability (S_f) in 1815-1822 relative to the pre-eruption period of 1799 to 1808. Following the comment, we have now clarified it in the caption of Figure 1. The revised Figure 1 is shown below:

Figure 1. Regional tree growth response to the 1815 Tambora eruption. Colors of markers on the geographic maps represent the change in growth (A) and stability (B) during a fixed period (1815 to 1822) compared to their pre-eruption period of 1789 to 1808 for each tree-ring chronology. Statistical plots show the probability distribution of the change of growth resilience (R_f , presented in bottom and left coordinate axes) and stability (S_f , presented in top and right coordinate axes) for evergreen needleleaf forests (ENF, green lines), deciduous needleleaf forests (DNF, orange red lines), deciduous broadleaf forests (DBF, cyan lines) and evergreen broadleaf forests (EBF, purple lines) within selected regions. Solid/dashed curves represent significant/non-significant difference ($P < 0.05$) between the mean growth and stability of the fixed period and pre-eruption period (i.e., 1799-1808) estimated using the Wilcoxon test. RAR, NEU, WCE, MED, MP, TIB, NWN, WUS, SWUS, CNA, ENA, SA and NZ represent Russian-Arctic, northern Europe, western and central Europe, the Mediterranean region, the Mongolian Plateau, the Tibetan Plateau, northwestern North America, the west coast of US, the Southwestern U.S., central North America, eastern North America, the southern Andes and New Zealand, respectively. Vertical dashed lines represent the median of

the distributions of R_f and S_f for each forest type within a given region.

[Comments] *Specific comments*

Lines 188-190: Why were negative exponential curves selected as a detrended approach taking into account that the simple exponent can be a reason of bias for the end of individual tree-ring series (Melvin, 2004)? Clarification is needed.

[Responses] Negative exponential curve is the most conservative standardization technique for detrending. Since the continuous two major eruptions have long-term effects on ecosystems, we chose to use this standardization technique. In this study, 3286 out of 3304 ring-width chronologies end after 1835, that is 20 years after the 1815 eruptions, therefore, the bias for the end of tree-ring series would not affect the reliability of the results. We also compared the SEA results with growth resilience based on RWI chronologies detrended by a 30-year cubic smoothing spline. Though there are some differences caused by different detrending methods, the results still suggested that the influence time can extend longer in mid- than in high-latitude regions. Following the suggestion, we have now clarified it in the Methods section: [Different detrending techniques may cause uncertainties when deriving the influence time of an extreme event. Since the two major consecutive eruptions were recorded to have long-term effects on ecosystems, we chose to use the most conservative standardization technique of fitting negative exponential curves which follows a model of tree growth to detrend tree-level series using the dplR package³⁵ (version 1.7.1) in R³⁶. Site-level, standard chronologies of RWI were averaged using a bi-weight robust mean.

For comparisons, a 30-year cubic smoothing spline was also used for detrending in order to verify the robustness of the derived influence time of volcanic forcing on tree growth against our choice of detrending method. This detrending method following an empirical model, removes 50% of variance at 30 years and leaves 99% of variance at 9.5 years (Cook & Peters, 1981). Through the lasting time showed a few years difference compared to detrending by fitting negative exponential curves, it still showed that the influence time can extend longer in mid-latitude regions than in high-latitude regions.]

[Comments] *Lines 194-202: How was the window length (left end – 1799-1808; right end – 1815-1722) selected?*

[Responses] In the revised version, we added Supplementary Figures 2 and 3 to present the stability of the results (please see above).

[Comments] *Equation 2: I would recommend the Authors to indicate that the R_i was used in Superposed Epoch Analysis and in Fig.2*

[Responses] Following the suggestion, we have now highlighted the use of growth resilience (R_i) in the “Method--Regional growth-anomaly assessment on growth resilience” section, and in the y-label and caption of Fig.2. In the “Method--Regional growth-anomaly assessment on growth resilience” section, we revised the context into: [In this study, we modified this

approach to evaluate the probability of the association between discrete volcanic eruptions and subsequent growth resilience (R_i) shifts on regional scales. We selected 13 regions on the basis of the spatial patterns of the site-level R_f and climatological consistency to examine the regional response of growth resilience to volcanic eruptions... We established regional series by joining a set of site-level R_i series within a selected region and applied the SEA to the joint regional series, comparing the 10 years before and 20 years after the first eruption year (1809).

Figure 2. Regional growth resilience (R_i) anomaly after the 1809 and 1815 volcanic eruptions within selected regions...]

[Comments] *Lines 215-217: Why was New Zealand not selected as a region for analysis taking into account its closest neighborhood with Tambora.*

[Responses] Following the suggestion, we have now added the analyses for New Zealand. Forests in New Zealand did not show significant growth reduction. This may be because the cooling effects were less strong in the Southern Hemisphere. We included the results of New Zealand into all the figures and added related discussion in the Discussion section: [Forests in New Zealand did not show significant growth reduction. This may be because the cooling effects were less strong in the Southern Hemisphere (Wilson et al., 2023), and surrounding ocean areas buffered the cooling effects.]

[Comments] *Lines 215-217: Why was the richest dendroregion (the central and southern West Coast of US) represented just by the southern part of the Colorado Plateau?*

[Responses] Following the suggestion, we have now added the analyses of forest resilience change in the west coast of the U.S. and central North America. Results were presented in all the figures and described in the Results and Discussion sections: [...In addition, western and central Europe, the Mongolian and Tibetan Plateaus, northwestern North America, the west coast of the U.S., the Southwestern U.S. (mainly mountainous regions in Arizona and New Mexico), central and eastern North America, and the southern Andes all experienced significant ($P < 0.05$) drops in growth for at least one forest functional type...

...Growth stability was also significantly ($P < 0.05$) reduced in the Russian Arctic, for larch forests in western and central Europe, for conifers along the west coast of the U.S. and in eastern North America, and for deciduous broadleaves (*Nothofagus*) in the southern Andes. By contrast, for evergreen needleleaf forests in northern Europe, in the Mediterranean region, and in New Zealand, as well as for deciduous broadleaf forests in western and central Europe, the growth stability significantly ($P < 0.05$) increased...

...The longest influence times (up to the year 1824) were found in the Mongolian and Tibetan Plateaus, the west coast of the U.S., mountainous regions of the Southwestern U.S., central North America, and the southern Andes....

...The longest influence time of volcanic cooling on tree growth was found in mid-latitudes in

the Northern Hemisphere; significantly reduced growth resilience extended up to 1824, that is, 9 years after the Tambora eruption. On the Mongolian and Tibetan Plateaus where vegetation growth is limited by both low growing-season temperature and precipitation, resilience decreased following the 1809 eruption and lasted until several years after the 1815 eruption; along the west coast of the U.S., the Southwestern U.S. mountainous regions, and central North America, vegetation growth is more limited by water availability than by temperature. Accordingly, we observed a rapid return of decreased growth to normal after the 1809 eruption, but a lasting resilience decreased followed the more explosive 1815 eruption...]

[Comments] *The reference list should be checked. The reference number 38 in the main text is not presented in the reference list.*

[Responses] Thank you for pointing out this mistake, we have now added the missed reference in the reference list: [37. Rao, M. P. et al. A double bootstrap approach to Superposed Epoch Analysis to evaluate response uncertainty. *Dendrochronologia*. 55, 119-124 (2019).]

References

- Dee, S. G. et al. No consistent ENSO response to volcanic forcing over the last millennium. *Science*. **367**, 1477 (2020).
- Liu, F. et al. Tropical volcanism enhanced the East Asian summer monsoon during the last millennium. *Nat. Commun.* **13**, (2022).
- Rao, M. P. et al. Seven centuries of reconstructed Brahmaputra River discharge demonstrate underestimated high discharge and flood hazard frequency. *Nat. Commun.* **11**, (2020).
- Wilson, N. et al. Impact of the Tambora volcanic eruption of 1815 on islands and relevance to future sunlight-blocking catastrophes. *Sci. Rep.*, **13**, 3649 (2023).

Reviewers' Comments:

Reviewer #1:

Remarks to the Author:

I have no remaining comments to add.

I think the paper could be accepted for publication.

Reviewer #2:

Remarks to the Author:

After the first round of peer-review the authors followed the all recommendations and revised the MS respectively.

I would recommend to publish the MS as it is.

Response to comments of Reviewer 1 and 2

REVIEWERS' COMMENTS

Reviewer #1 (Remarks to the Author):

I have no remaining comments to add.
I think the paper could be accepted for publication.

Reviewer #2 (Remarks to the Author):

After the first round of peer-review the authors followed the all recommendations and revised the MS respectively.
I would recommend to publish the MS as it is.

[Responses] We appreciate the reviewers' positive assessment of the study. We also very much appreciate their insightful comments, which have helped us to improve the manuscript considerably.